# Elucidation of trophic interactions in an unusual single-cell nitrogen-fixing symbiosis using metabolic modeling

Debolina Sarkar[1], Marine Landa[2], Anindita Bandyopadhyay[3], Himadri B. Pakrasi[3], Jonathan P. Zehr[2], Costas D. Maranas[1]*

**1** Department of Chemical Engineering, Pennsylvania State University, University Park, Pennsylvania, United States of America, **2** Department of Ocean Sciences, University of California, Santa Cruz, California, United States of America, **3** Department of Biology, Washington University, St. Louis, Missouri, United States of America

* costas@psu.edu

**Data Availability Statement:** All codes and data files can be found at https://github.com/

## Abstract

Marine nitrogen-fixing microorganisms are an important source of fixed nitrogen in oceanic ecosystems. The colonial cyanobacterium *Trichodesmium* and diatom symbionts were thought to be the primary contributors to oceanic $N_2$ fixation until the discovery of the unusual uncultivated symbiotic cyanobacterium UCYN-A (*Candidatus Atelocyanobacterium thalassa*). UCYN-A has atypical metabolic characteristics lacking the oxygen-evolving photosystem II, the tricarboxylic acid cycle, the carbon-fixation enzyme RuBisCo and *de novo* biosynthetic pathways for a number of amino acids and nucleotides. Therefore, it is obligately symbiotic with its single-celled haptophyte algal host. UCYN-A receives fixed carbon from its host and returns fixed nitrogen, but further insights into this symbiosis are precluded by both UCYN-A and its host being uncultured. In order to investigate how this syntrophy is coordinated, we reconstructed bottom-up genome-scale metabolic models of UCYN-A and its algal partner to explore possible trophic scenarios, focusing on nitrogen fixation and biomass synthesis. Since both partners are uncultivated and only the genome sequence of UCYN-A is available, we used the phylogenetically related *Chrysochromulina tobin* as a proxy for the host. Through the use of flux balance analysis (FBA), we determined the minimal set of metabolites and biochemical functions that must be shared between the two organisms to ensure viability and growth. We quantitatively investigated the metabolic characteristics that facilitate daytime $N_2$ fixation in UCYN-A and possible oxygen-scavenging mechanisms needed to create an anaerobic environment to allow nitrogenase to function. This is the first application of an FBA framework to examine the tight metabolic coupling between uncultivated microbes in marine symbiotic communities and provides a roadmap for future efforts focusing on such specialized systems.

maranasgroup/UCYN-A-symbiosis-metabolic-modeling.

**Funding:** This research received funding from the Center for Bioenergy Innovation (CBI) (DE-AC05-00OR22725) to C.D.M, Department of Energy (DESC0019386) to C.D.M. and H.B.P, Gordon and Betty Moore Foundation (GBMF5760) and National Science Foundation (MCB 1933660) to H.B.P, Simons Foundation (Award ID 545171) to J.Z. The funders had no role in study design, data collection and analysis, decision to publish, or preparation of the manuscript.

**Competing interests:** The authors have declared that no competing interests exist.

## Author summary

Reduction of dinitrogen gas to biologically useful forms via nitrogen fixation is a key component of the biogeochemical cycle. In the marine environment, the cyanobacteria UCYN-A (*Candidatus Atelocyanobacterium thalassa*) has been found to be a primary contributor to biological nitrogen fixation at a global scale. UCYN-A exhibits a highly streamlined genome which lacks genes coding for essential cyanobacterial processes such as the energy-generating TCA cycle, oxygen-producing photosystem II, the carbon-fixing RuBisCo and *de novo* production pathways for numerous amino acids and nucleotides. Thus, it exists in a symbiosis with unicellular planktonic algae where it exchanges fixed nitrogen for fixed carbon with its host. However, both UCYN-A and its symbiotic partner remain uncultured under laboratory conditions. This necessitates implementing a computational approach to glean insights into UCYN-A's unique physiology and metabolic processes governing the symbiotic association. To this end, we develop a constraints-based framework that infers all possible trophic scenarios consistent with the observed data. Possible mechanisms employed by UCYN-A to accommodate diazotrophy with daytime carbon fixation by the host (i.e., two mutually incompatible processes) are also elucidated. We envision that the developed framework using UCYN-A and its algal host will be used as a roadmap and motivate the study of similarly unique microbial systems in the future.

## Introduction

The reduction of atmospheric nitrogen to ammonia is an energy intensive process that is necessary to supply nitrogen to terrestrial and aquatic ecosystems. First discovered in 1880 by Hellriegel and Wilfarth [1] in legumes and cereals, biological $N_2$ fixation is performed by free-living and symbiotic Archaea and Bacteria inhabiting a variety of habitats, including soils, rice fields, lacustrine waters, and the ocean [2]. These associations range from free-living diazotrophs, to intercellular endophytic associations, and endosymbiosis. The underlying molecular mechanisms are equally diverse, with legumes forming nodules to host rhizobium and filamentous cyanobacteria developing specialized cells (called heterocysts) to allow spatial separation of oxygen-sensitive nitrogen fixation and oxygen-evolving photosynthesis.

Oceanic $N_2$ fixation has garnered interest in recent years because it was suggested that there was an imbalance in the oceanic fixed N budget [3]. Until very recently the filamentous, colony-forming cyanobacterium *Trichodesmium* and symbionts of diatoms such as *Richelia* were believed to be the major oceanic diazotrophs [4]. However, the use of polymerase chain reaction (PCR) to amplify the *nifH* gene (which encodes the iron subunit of nitrogenase) [5] revealed the presence of unicellular diazotrophic cyanobacteria, and led to the discovery of the unusual UCYN-A group (*Candidatus Atelocyanobacterium thalassa*). UCYN-A is most closely related to the unicellular free-living *C. watsonii* and *Cyanothece* sp. ATCC 51142 [6], and is widely distributed in the ocean, fixing $N_2$ at rates equal or greater than those of *Trichodesmium*. This discovery also expanded the geographic range of oceanic $N_2$-fixation to colder and more nutrient-rich areas [7].

UCYN-A forms a metabolic partnership with a single-celled haptophyte belonging to the *Braarudosphaera bigelowii* clade [8], which remains uncultivated and only partially sequenced [9]. There are several UCYN-A lineages with high degree of specificity between symbiotic partners and the reductive evolution of UCYN-A genomes [10], as well as experimental observations of endosymbiosis between one of the UCYN-A lineages and *B. bigelowii* [11]. An

obligatory dependence of UCYN-A on its host was hypothesized and supported by multiple additional lines of evidence such as the strong coupling in carbon and nitrogen sharing between partner cells [12] and the absence of observations of free-living hosts [13]. Visualization of the symbiosis using nanometer scale secondary ion mass spectrometry showed that carbon is fixed by the host and transferred to UCYN-A, which in turn fixes nitrogen and supplies it to the host [8]. The proposed symbiosis hypothesis is further strengthened by the radical genome reduction in UCYN-A which lacks $O_2$-evolving PSII, enzymes required for carbon fixation, the tricarboxylic acid cycle, and biosynthetic pathways for a majority of amino acids and nucleotides, making it a highly unusual cyanobacterium. This implies that many essential metabolic functions must be supplemented by the host. However, the identity of the metabolites that are exchanged and the resulting metabolic interactions remain unknown, because the symbiosis (and individual partners) are yet to be cultured under laboratory conditions. Understanding the mechanisms involved in the UCYN-A symbiosis is also important as it is akin to the early stages of endosymbiosis and the evolution of plastids, offering an exemplar to study the evolution of a hypothetical $N_2$-fixing organelle or "nitroplast". A similar example can be found in the 'spheroid bodies' observed in diatoms from the family *Rhopalodiaceae*–these lack genes for both PSI and PSII, and have an incomplete TCA cycle, but possess complete biosynthetic pathways for amino acids, nucleotides, and cofactors in a genome that is almost twice the size of UCYN-A (1.44 Mbp vs 2.79 Mbp) [14]. Thus, despite the strong coevolutionary histories observed in a majority of symbioses in nature, only a few exhibit the loss of individual autonomies. Herein lies the distinction that singles out the UCYN-A and haptophytic host unicellular association, enabling us to study the evolutionary transition between symbiotic partnerships and new, integrated organisms.

In this work, we used flux balance analyses (FBA) to further investigations into this unique symbiosis between unicellular microbes by exploring the potential metabolic interdependencies between UCYN-A and its haptophyte host. To this end, genome-scale metabolic reconstructions were created for both organisms, using the genome sequence of *Chrysochromulina tobin* as a proxy for the host. A set of essential biomass precursors was assembled for UCYN-A based on existing metabolic reconstructions of model $N_2$-fixing (*Cyanothece* sp. ATCC 51142 [15]) and minimalistic (*P. marinus* [16]) cyanobacteria. By assessing both host and UCYN-A metabolisms together, we determined a minimal set of metabolites and alternates needed from the host to produce all UCYN-A biomass precursors and the specific roles played by the two partners to facilitate symbiosis. We found that a minimum of 28 metabolites must be provided by the host to enable UCYN-A growth, out of which twenty metabolites are essential with alternative choices for the remaining eight. Some of the predicted metabolite exchange patterns (such as transferring fixed nitrogen as glutamine or ammonia) is akin to the exchange of metabolites between heterocysts and vegetative cells of heterocystous cyanobacteria. However, it would be naïve to classify UCYN-A as a heterocyst as the underlying metabolic capabilities are vastly different and lack of specialized cellular substructures. For example, unlike heterocysts which preferably import sucrose [17], UCYN-A must rely on alternate carbon substrates as it does not possess a TCA cycle. Heterocysts are further protected from nitrogenase poisoning by oxygen released from the vegetative cells due to the thick cell wall created during differentiation, which has not been observed in UCYN-A.

To this end, we further explored the possible mechanisms that enable UCYN-A to fix nitrogen in the daytime while avoiding oxygen inactivation of nitrogenase. By modeling the symbiosis between UCYN-A and its prymnesiophyte host, we can thus identify the minimum constraints required to facilitate single-cell symbiosis.

## Results

### UCYN-A and host genome scale metabolic model reconstructions

Annotated reactions from the genomes of two model cyanobacterial strains were mapped to the UCYN-A genome to generate a genome-scale metabolic model (GSM). A GSM is a mathematical representation of an organism's biochemistry, containing information on all known metabolic reactions, the genes encoding each enzyme and biomass constituents and proportions. The same workflow was used to generate a metabolic model of the host, mapping reactions directly from existing GSMs of four phototrophs (see Methods). We chose the haptophyte *C. tobin* as representative of the host since the genomes of the known UCYN-A haptophyte partners [8] are only partially sequenced. The two metabolic networks were linked by adding transport reactions that could ferry metabolites between the symbionts. By performing FBA for both models simultaneously, gaps in UCYN-A metabolism were identified and compensation scenarios offered by the host were constructed. Biomass composition from *C. reinhardtii* was used for the host GSM, and UCYN-A's biomass precursors were adapted from *Cyanothece* sp. ATCC 51142 and *P. marinus* (see Methods). FBA was carried out by requiring that each UCYN-A biomass precursor was produced at a minimal level (as the exact biomass composition is unknown) while minimizing the number of distinct metabolites exchanged between them (see minTransfers in Methods) [18,19]. This modeling posture implies that metabolite exchange happens on a "only when required" basis while both organisms are sequestering metabolic precursors originating from carbon and nitrogen fixation into biomass. All simulations were performed under phototrophic conditions. A total of 100 alternate solutions were generated to explore various alternative metabolite exchange scenarios. Carbon was supplied as $CO_2$ and nitrogen as molecular $N_2$ to the system. The trophic scenarios were further constrained using experimentally-measured rates of total carbon and nitrogen exchange, wherein at most 17% of the fixed carbon was allowed to be transferred from *C. tobin* to UCYN-A and up to 95% of fixed nitrogen from UCYN-A back to the algal host [8]. However, no constraints were imposed on specific reactions associated with carbon or nitrogen fixation so as not to bias results towards a particular phenotype.

### Metabolites transferred from the host to the symbiont

As the biochemical composition of UCYN-A is uncharacterized, a trade-off analysis of growth rates with metabolite sharing in the symbiotic system is prohibitive. Thus, we determined the minimal metabolite set that must be exchanged between the two partners to enable UCYN-A growth. This can be considered to be the lower bound for metabolite sharing in the symbiosis, below which the system will collapse. As expected, we found that UCYN-A's primary role in the symbiosis is to fix nitrogen. Part of the fixed nitrogen is transferred to the host as a combination of ammonia, alanine, and glycine (Fig 1). Nitrogen transfer via alanine and glycine requires the import of carbon substrates from the host as their synthesis in UCYN-A proceeds via the amination of pyruvate by alanine dehydrogenase. The pyruvate substrate must either be imported from the host or synthesized via glycolysis by importing a further upstream precursor. The produced alanine can be exported to the host to be readily incorporated into proteins, or down-converted to glycine via the alanine-glyoxylate aminotransferase while importing glyoxylate from the host. The imported nitrogenous compound is subsequently utilized by the host to synthesize amino acids, nucleotides, and pigments such as carotenes and xanthins. The scenario wherein the host can retrieve fixed nitrogen from the environment was also explored (see Fig 1 in S1 Text); however a recent study [20] determined that even in dissolved nitrogen replete areas, the host meets little of its nitrogen demand via ammonium

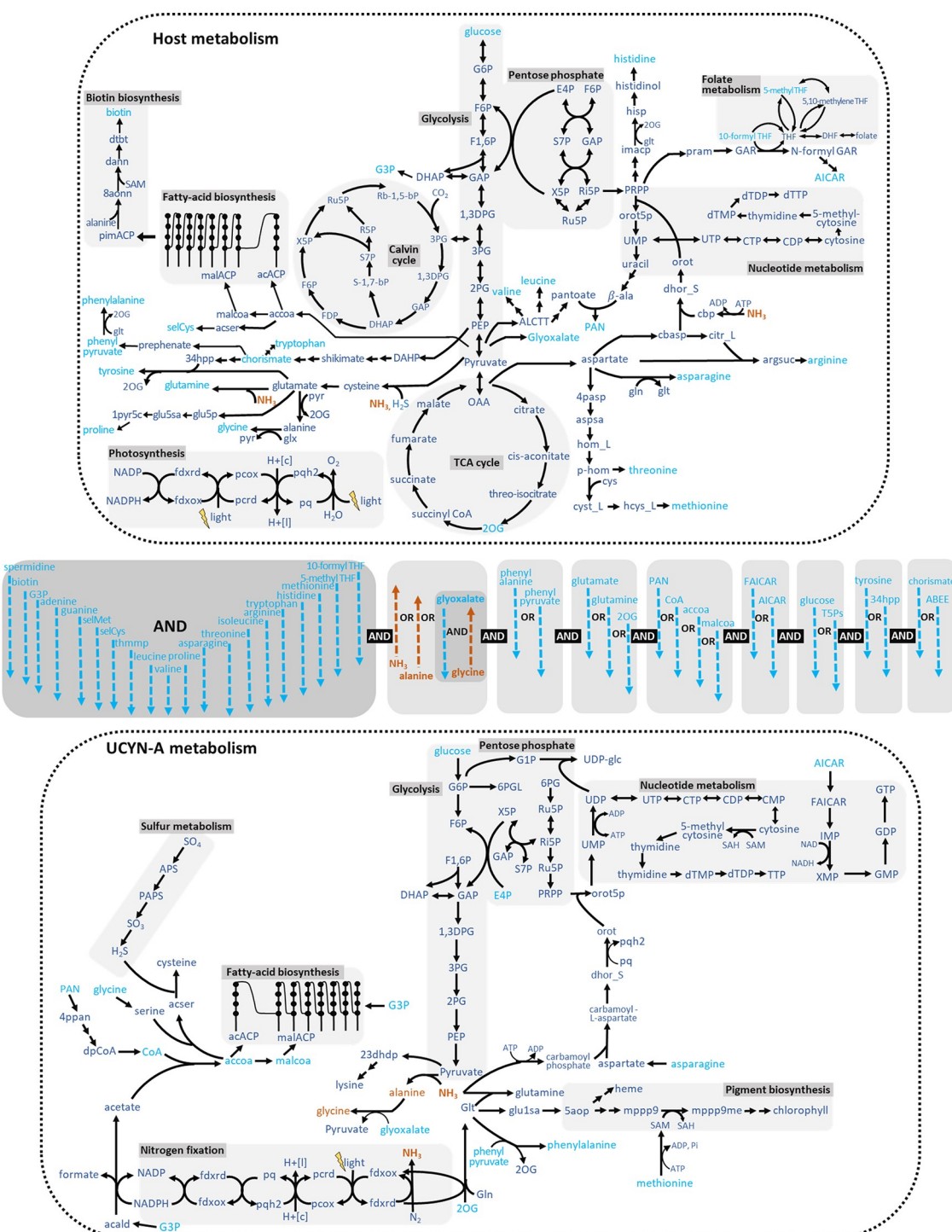

**Fig 1. Summary of the metabolite predicted exchanges between UCYN-A and its host.** Metabolic pathway diagram showing symbiosis between the algal host (top) and UCYN-A (bottom). Metabolite transfers are grouped using Boolean statements wherein metabolites which must be transferred simultaneously to realize biomass synthesis are shown using an AND statement and metabolites which represent alternate trophic scenarios are shown using an OR statement. Metabolites transferred from the host are marked in cyan while metabolites transferred from UCYN-A are shown in orange. Host metabolites that must necessarily be transferred have been grouped together using an AND statement (left-most block, shown in dark grey). Metabolites IDs conforming to the BiGG database have been used.

uptake. This further strengthens the predicted metabolic roles of the two partners, wherein the host provides fixed carbon and UCYN-A provides fixed nitrogen.

At least 28 metabolites must be transferred from the host to UCYN-A to enable growth, out of which 20 were present in all trophic scenarios with no alternates (Fig 1 and S1 Table). These included ten amino acids, purines adenine and guanine, vitamins biotin (B7), folates (B9) and thiamine (B1) as well as glycerol-3-phosphate (G3P) for fatty acid synthesis (see Fig 1, metabolites linked with AND statements). Fixed carbon can be transferred as glucose but alternate solutions include transfer via acetaldehyde or triose phosphates. Metabolic modeling revealed a number of alternate trophic scenarios involving either the direct transfer of terminal biomass precursors or upstream intermediates of their biosynthetic pathways. Fig 2 illustrates these alternatives for amino acids lysine, alanine, serine, cysteine, glutamine, glutamate, phenylalanine, tyrosine, aspartate and glycine for which UCYN-A possesses incomplete metabolic pathways. Glutamate synthesis proceeds via ferredoxin-dependent glutamate synthase (GOGAT) which is coupled to glutamine synthetase (GS). The net reaction incorporates $NH_3$ (produced during $N_2$-fixation) to 2-oxoglutarate (2-OG) at the expense of ATP and reducing power. The diel expression patterns of genes UCYN_11890 (encoding for GS) and UCYN_03690 (encoding for GOGAT) show significant correlation with UCYN_06160 (encoding for the alpha subunit of nitrogenase) (pearson's correlation coefficients 0.82 and 0.91, p-values 0.013 and 0.0019 respectively) [21]. This indicates that the scenario wherein fixed ammonia is assimilated into 2-OG via the GS-GOGAT cycle and transferred as glutamine/glutamate to the host is indeed feasible in UCYN-A.

Subsequent transamination of host-produced phenylpyruvate with glutamate yields phenylalanine. Thus, either phenylalanine or phenylpyruvate must be imported from the host as UCYN-A possesses the gene encoding for phenylalanine transaminase but lacks the pathway producing phenylpyruvate. Similar alternate trophic scenarios are predicted for tyrosine whose synthesis proceeds via tyrosine aminotransferase. UCYN-A must either directly import tyrosine or the intermediate p-hydroxy phenylpyruvate.

The synthesis of the remaining glucogenic amino acids, (*i.e.*, alanine, serine, and cysteine) is enabled by the import of fixed carbon from the host (shown as erythrose-4-phosphate in Fig 2). Host-derived erythrose-4-phosphate (E4P) can be converted into pyruvate, then alanine and finally glycine (as described above) using glyoxylate provided by the host. E4P is first converted to pyruvate via lower glycolysis. Pyruvate is transaminated via alanine dehydrogenase which incorporates ammonia obtained from nitrogen fixation to produce alanine. Alanine-glyoxylate aminotransferase can generate glycine using glyoxalate obtained from the host. Serine hydroxymethyltransferase then converts glycine to serine. L-serine acetyltransferase can transfer the acetyl group from acetyl-CoA to serine produce O-acetylserine, which upon condensation with sulfide yields cysteine. Sulfide is a product of assimilatory sulfate reduction, which is notably one of the few metabolic pathways to be conserved in its entirety in UCYN-A [22]. Under this minimal trophic scenario, UCYN-A imports ten amino acids (*i.e.*, leucine, proline, valine, methionine, isoleucine, histidine, asparagine, tryptophan, arginine, and threonine) and synthesizes the rest using carbon substrates E4P, glyoxylate, and 2-oxoglutarate from the host. Putative transporters for 23 metabolites could be identified using the UCYN-A genome annotation (S3 Table).

Our computational analysis also helped identify metabolic dependencies between distal biochemical pathways. For example, the import of asparagine by UCYN-A is required for the production of nucleotides and associated sugars such as UDP-glucose and dTDP-rhamnose (Fig 1). Methionine influx was also identified as necessary for UCYN-A growth, being a precursor of the methyl-donating cofactor S-adenosyl methionine (SAM). SAM is a ubiquitous cofactor in the cell, participating as the methylation agent for a number of reactions across pathways

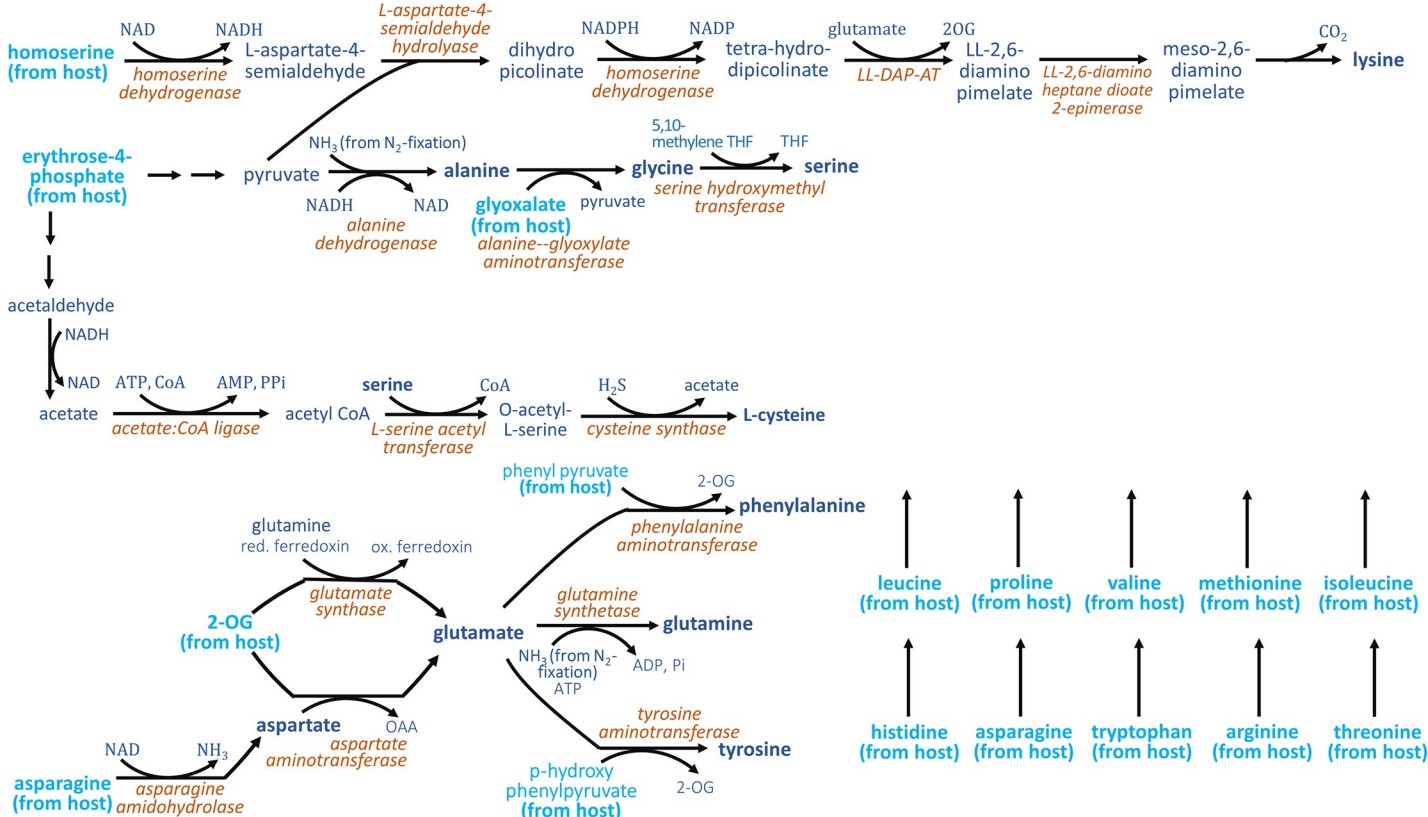

**Fig 2. Amino acid biosynthesis in UCYN-A.** Overview of amino acid biosynthesis and predicted import in UCYN-A. Metabolites synthesized by UCYN are shown in blue, and metabolites transferred from the host in cyan. Enzyme names are shown in orange.

such as nucleotide, and pigment biosynthesis (see Fig 1). For example, SAM acts as the methylating agent in the conversion of Mg-protoporphyrin IX to Mg-Protoporphyrin IX 13-monomethyl ester, which is a chlorophyll precursor (see Fig 1). In the nucleotide biosynthesis pathway, SAM is required to produce 5-methylcytosine from cytosine, which then gives rise to thymine and thymidine. This highlights the importance of adapting a network view of metabolism for elucidating non-trivial interdependencies, which can often be missed in a pathway-wise analysis.

## Symbiosis enables daytime $N_2$-fixation in UCYN-A

UCYN-A exhibits high nitrogenase activity in the daytime which is unusual for a cyanobacteria lacking heterocysts [23,24]. Metabolic modeling yields results consistent with the hypothesis that nitrogen fixation depends on the supply of fixed carbon from the host. UCYN-A reactions involved in diazotrophy (either directly such as nitrogenase or facilitating it by producing reductants and ATP) have high flux control coefficients for both host biomass production and $N_2$ fixation (see Table 2 in S1 Text). In this scenario, carbon substrates imported via carbohydrate porins or ABC transporters are used to generate reductants via oxidoreductases which fuel nitrogenase (Fig 3). The generated NADPH transfers electrons to ferredoxin via ferredoxin:NADP reductase (FNR) to reduce the plastoquinone pool via cyclic electron flow (FQR), cytochrome b6/f complex, and PSI. This prediction is consistent with high FNR transcripts observed during the day in UCYN-A [21]. Although NADPH can be generated by FNR

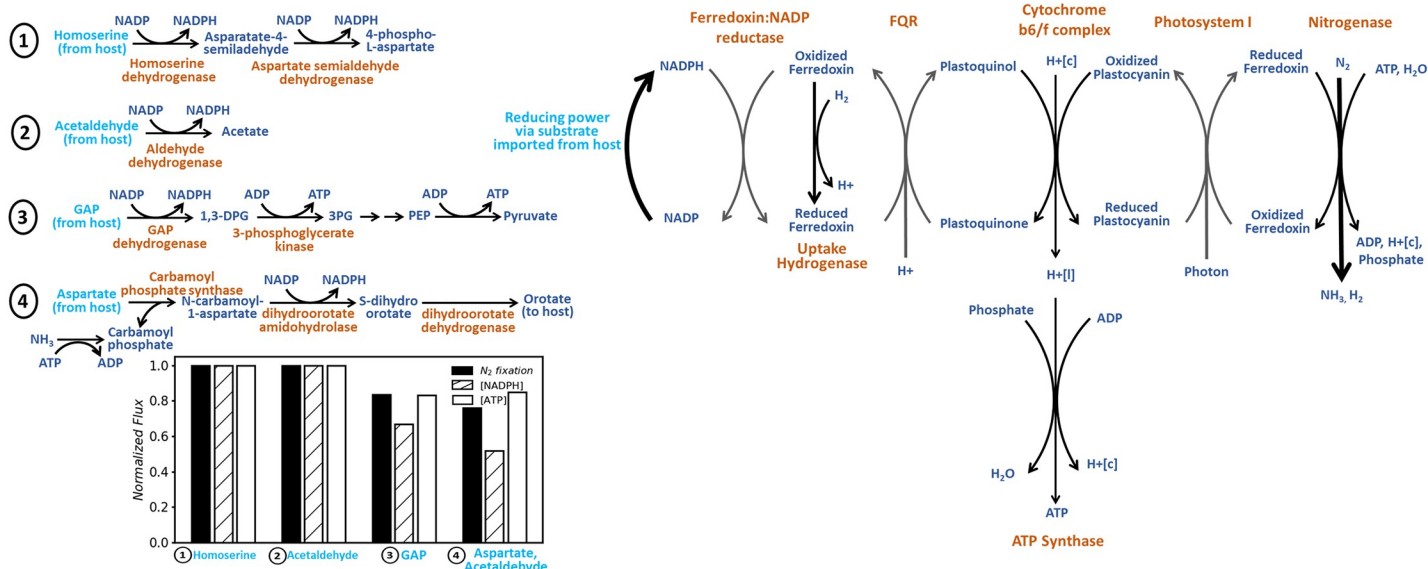

**Fig 3. Nitrogen fixation in UCYN-A.** Reactions associated with nitrogen fixation in UCYN-A and the facilitating substrates imported from the host. Substrates (shown in cyan) have been ranked in order of their N$_2$-fixation yield (mmol N$_2$ fixed/mmol CO$_2$ uptake). The bar chart shows nitrogen fixation, total ATP yield, and total NADPH yield flux for four alternative substrates, normalized by the maximal value observed across substrates. Metabolites are shown in blue and enzymes in orange in the pathway diagram (right).

using reduced ferredoxin, our simulations indicate that FNR functions in reverse to instead reduce ferredoxin with NADPH generated by carbohydrate oxidation, akin to that seen in heterocysts [25,26]. The predicted optimal flux distributions further suggest that ATP is generated by ATP synthase using the proton gradient created by the cytochrome b6/f complex. The hydrogen produced during nitrogen fixation can be recycled using uptake hydrogenase (encoded by genes UCYN_00710 and UCYN_00690) or off-gassed to the environment. Generally, N$_2$-fixing cyanobacteria show little net H$_2$ production due to the efficient recycling by uptake hydrogenase. This provides additional reducing power for diazotrophy and other cellular processes.

In order to determine the probable carbon substrates imported by UCYN-A, we calculated the nitrogen fixation yield (mmol N$_2$ fixed/mmol CO$_2$ uptake by the host) associated with each imported metabolite candidate (see Fig 3 and S1 Table). This was calculated by maximizing flux through the nitrogen fixation reaction while maintaining the number of metabolites exchanged at a minimum (see problem maxFixation in Methods). As many as 100 alternate import scenarios were generated (S1 Table). Results indicate that the ranking of the exchanged metabolites is determined by the net ATP and NADPH that they can provide (Fig 3). The highest nitrogen fixation yield is associated with the import of homoserine and acetaldehyde from the host, followed by glycolytic intermediates such as glyceraldehyde-3-phosphate (GAP), fructose-1,6-bisphosphate (F-1,6-P), and dihydroxyacetone phosphate (DHAP) and downstream products such as aspartate. A similar observation for a cell-free system derived from heterocysts wherein substrates GAP, DHAP, and F-1,6-P supported high nitrogenase activity [27]. This implies that priming of nitrogen fixation in UCYN-A is similar to that of heterocystous cyanobacteria.

## Evaluating oxygen scavenging mechanisms

Even though UCYN-A metabolism is akin to heterocystous cyanobacteria wherein nitrogenase activity is dependent on substrates imported from the host [28], UCYN-A lacks the

characteristic double-layered cell envelope that prevents oxygen entry. $N_2$-fixation is a strictly anaerobic process as the iron-sulfur clusters in nitrogenase become irreversibly oxidized and thus rendered catalytically inactive by molecular oxygen. Although UCYN-A avoids oxygen production by not splitting water using photosystem II, the photosynthetic host alga does evolve oxygen. Furthermore, nitrogen and oxygen molecules have similar sizes (1.09 Å vs 1.11 Å interatomic distances) leading to similar permeabilities through the plasma membrane. Thus, UCYN-A must consume molecular oxygen at high rates in order to maintain anoxia in the vicinity of the enzyme. Cyanobacteria have evolved three major mechanisms for doing so–photorespiration using the oxygenase activity of RuBisCo, aerobic (cytochrome-dependent) respiration, and photocatalyzed reduction of oxygen to water in PSI ('Mehler reaction') [29]. Only the last two are available to UCYN-A due to the absence of RuBisCo [21]. Genes necessary for cytochrome-dependent respiration (reaction ID CYO1b2pp_syn) are UCYN_12280, UCYN_12290, UCYN_12300, and UCYN_02310, and for the Mehler reaction (reaction ID R_MEHLER) is UCYN_04350.

We systematically investigated the preferred oxygen scavenging mechanism by determining the maximum theoretical nitrogen fixation flux while relying on either cytochrome-dependent respiration or the Mehler reaction to consume a fixed amount of oxygen (see maxFixation in Methods). The respective stoichiometries for the overall reactions are:

**Cytochrome-dependent resp:** $(4)H^+$ + (2) Reduced plastocyanin + $(\frac{1}{2})$ $O_2$ = > $(2)H^+[l]$ + (2) Oxidized plastocyanin + $H_2O$

**Mehler reaction:** (2) $H^+$ + (2) Reduced ferredoxin + $O_2$ = > (2) Oxidized ferredoxin + $H_2O_2$

Cytochrome c oxidase directly transfers electrons from plastocyanin to oxygen while the Mehler reaction uses ferredoxin as a carrier. Model predictions indicate that using the Mehler reaction to create anoxic conditions can support an approximately 15% higher nitrogen fixation flux compared to cytochrome c oxidase for the same amount of scavenged oxygen. This is because for the same amount of reducing power (provided as photosynthates by the host), the net ATP production by the Mehler reaction is ~15% greater (Fig 4). This prediction is consistent with earlier observations reporting higher oxygen consumption via the Mehler reaction under light in diazotrophic cyanobacteria for both laboratory [30] and field conditions [31]. As PSI is usually fueled by electrons obtained by splitting water by PSII, Mehler activity is generally assumed to only consume photosynthetically produced oxygen [32]. However, in cyanobacteria such as *Trichodesmium* the arrangement of the photosynthetic and respiratory electron transport chains permits electrons derived from NAD(P)H to enter the photosynthetic electron transport chain and reduce PSI [33]. Modeling results herein support a similar mechanism at work that enables the Mehler reaction to proceed in UCYN-A without the presence of any PSII activity (Fig 4). Note that UCYN-A possesses the antioxidants required to reduce the peroxide by-product of the Mehler reaction, offering a mechanism for amending cell toxicity. Furthermore, higher transcript levels have been reported for the enzymes superoxide dismutase (*sod1*) and two peroxiredoxins (*prxR*) during the day in UCYN-A [21].

## Discussion

Symbiotic interactions are prevalent in natural systems and their onset was critical for the evolution of eukaryotic life. Such interactions can range from those between multicellular plants and unicellular microbes to ones between unicellular organisms. Recent studies show that the nature of these interactions vary considerably and deploy vastly different exchange strategies to maintain symbiosis. The focus of this study was to explore the metabolic basis of the interactions between the single-celled cyanobacterium UCYN-A and its prymnesiophytic microalgal

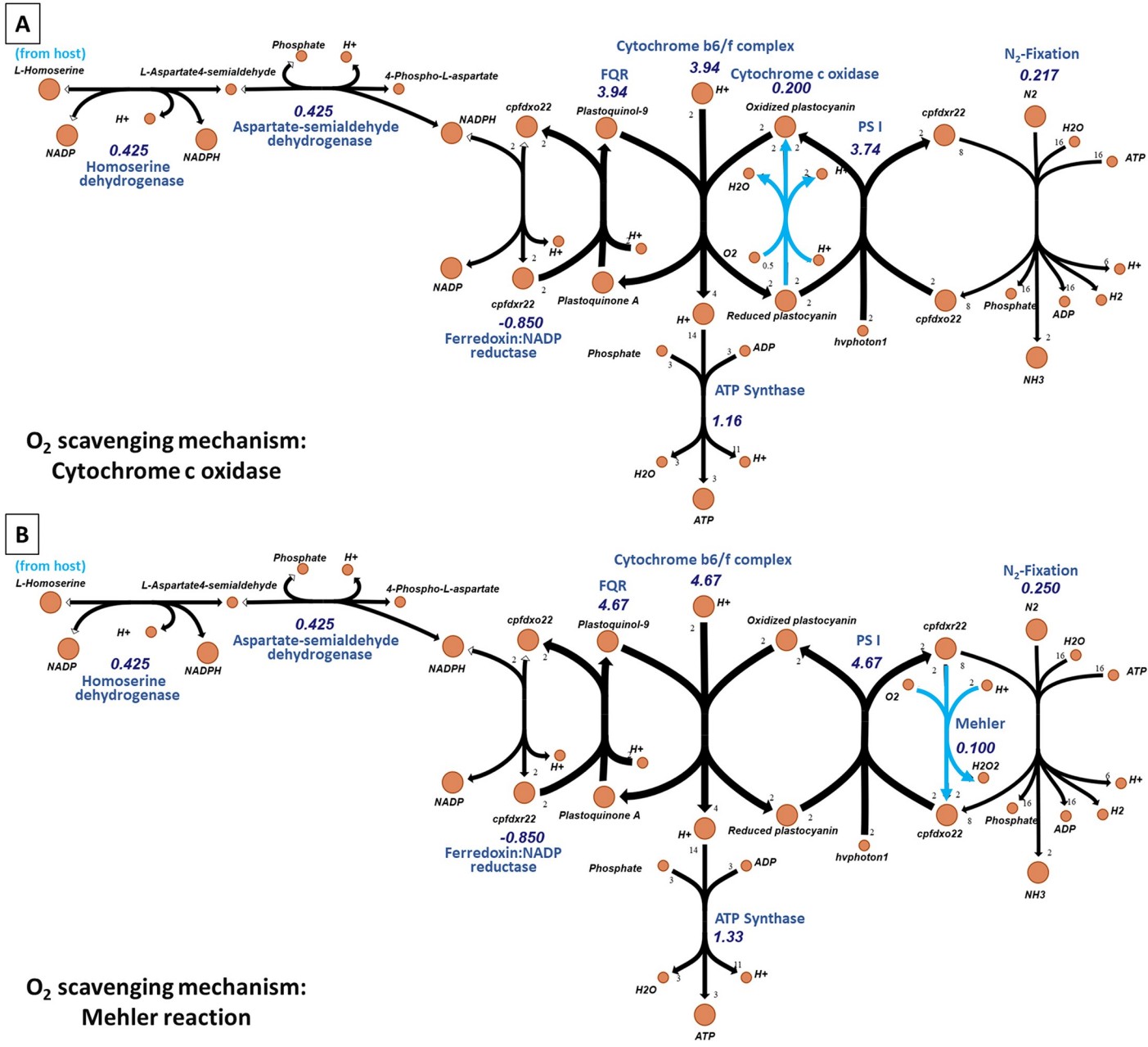

**Fig 4. Calculated flux distributions under nitrogen fixation comparing two oxygen-scavenging mechanisms.** Model-predicted reaction fluxes (values in mmol gDW$^{-1}$ hr$^{-1}$) when maximizing nitrogen fixation flux and employing (A) cytochrome-dependent respiration, and (B) the Mehler reaction to consume oxygen at 0.1 mmol$^{-1}$ gDW hr$^{-1}$. The competing mechanisms are shown in cyan. Metabolites are shown as nodes (in orange) and reactions as directed edges connecting them. The thickness of an edge corresponds to the flux through it. Reaction fluxes were computed using a basis of 10 mmol gDW$^{-1}$ hr$^{-1}$ of supplied $CO_2$.

host, the only known instance of a nitrogen-fixing symbiosis with a haptophyte. Experimental investigations into this system have so far remained elusive due to difficulties in culturing UCYN-A and/or its host. This has rendered computational studies on metabolic models an essential tool for deciphering possible trophic scenarios.

UCYN-A has numberous incomplete anabolic pathways and lacks essential genes encoding PSII, RuBisCO and the TCA cycle in its entirety. Therefore, it was suggested [11] that it forms

a symbiosis with its algal host. However, the exact nature of metabolic exchanges between the two organisms is still unknown. By modeling their respective metabolic capabilities using genome-scale metabolic reconstructions, we explored trophic scenarios required for the obligatory symbiosis. We found that the primary role of UCYN-A was to provide fixed nitrogen to its phototrophic host, which in turn provides fixed carbon to UCYN-A. Nitrogen transfer could occur directly as ammonia or through amino acids alanine and/or glycine. However, amino acid-based nitrogen transfer requires influx from the host of a glycolytic precursor. UCYN-A has a radically streamlined genome requiring the transfer of at least 28 distinct metabolites to enable growth. For 20 out of 28 metabolites no alternatives were found, implying the obligatory nature of their exchange. These include ten amino acids, the purines adenine and guanine, a number of vitamins, and carbon intermediates such as glycerol-3-phosphate and glycolytic intermediates. Flux balance analysis suggested that the import of either final precursors or metabolic intermediates can compensate for the incomplete anabolic pathways. The suggested trophic scenarios can inform UCYN-A growth under laboratory conditions by pinpointing components to include in the growth medium.

Apart from the extreme metabolic streamlining, another unique attribute of the UCYN-A symbiosis is that the nitrogenase activity has surprisingly shifted to daytime [23,24,34]. Daytime nitrogen fixers must physically separate the two processes by forming heterocysts while related diazotrophic cyanobacteria such as *Crocosphaera* and *Cyanothece* fix nitrogen during the night to protect nitrogenase against oxygen evolved during daytime photosynthesis. We found that this surprising timing of nitrogen fixation in UCYN-A can be explained by suggesting modified roles for a number of metabolic processes in both organisms. One such modified role is that UCYN metabolism is primed towards utilizing host photosynthates (*i.e.*, acetaldehyde, GAP, F-1,6-P, and DHAP) for generating reductants via oxidoreductases and light for ATP generation by the ATP synthase using the proton gradient generated by the cytochrome b6/f complex and PSI. Daytime nitrogen fixation also implies that the symbiosis has developed strategies to prevent inhibition of nitrogenase by the oxygen evolved during host photosynthesis. We evaluated two oxygen scavenging mechanisms available (*i.e.*, cytochrome-dependent respiration vs. the Mehler reaction) and found that the Mehler reaction is associated with a higher theoretical diazotrophic efficiency due to a higher ATP production flux. Thus, the UCYN-A metabolism appears to be optimized to support maximal nitrogen fixation flux alluding that this symbiosis is as close to being a functional 'nitroplast' as any observed till date.

The relative paucity of experimental data for the studied symbiotic system necessitates the adoption of a modeling/computational approach to infer all possible trophic scenarios consistent with the observables. However, the same dearth of data precludes the identification of a unique solution for the intra- and inter- organismal metabolic fluxes. We envision that the developed formalism will be successively used to prune away alternative trophic scenarios and move towards a unique solution as more data become available for this system and others.

## Methods

### UCYN-A and algal host metabolic reconstruction

We first constructed a UCYN-A draft model by aggregating reactions from the RAST-annotated [35] genome sequence assembled by Tripp et al. [22]. Although uncultured, the UCYN-A genome sequence was first assembled into one scaffold containing gaps of known lengths, and then closed using a combination of contig pooling and PCR by Tripp and coworkers [22]. Metabolic reactions were mapped from *Cyanothece* sp. ATCC 51142 [15] and *Prochlorococcus marinus* [16]. *Cyanothece* sp. ATCC 51142 was chosen due to its similarity to the UCYN-A genome, especially the nitrogenase *nif* gene cluster [36]. *P. marinus* has the smallest

genome of a photosynthetic organism known to date and lacks genes involved in functions that are conserved in cyanobacteria, such as photosynthesis, DNA repair, and solute uptake [37]. As UCYN-A also possesses a minimal genome, *P. marinus*' genome reduction was the primary motivation behind its selection as a scaffold organism. Gene homologs between these organisms and UCYN-A were found using a bidirectional protein BLAST. A requirement of mutually-best hits was imposed alongside an e-value cutoff of $10^{-30}$ for every match, so as to avoid spurious hits [38]. Next, reaction sharing between UCYN-A and the reference organisms was determined by evaluating the Boolean logic implied by each gene-protein-reaction relationship in *Cyanothece* and *P. marinus*. A reaction was transferred to UCYN-A only if it was found to possess the homologs required to satisfy the logic and thus encode the corresponding protein.

A similar procedure was implemented while constructing the algal host GSM. We used the haptophyte *Chrysochromulina tobin* [39] as the representative host as the genomes of the known UCYN-A symbiotic partners *B. bigelowii* and *C. parkeae* [8] are only partially sequenced. This includes segments of their ribosomal DNA for phylogenetic studies [8,40,41], thus precluding including any host genes or metabolic functions in the current reconstruction. Metabolic reactions were taken from the *C. tobin* annotated genome sequence and also mapped from four existing GSMs of phototrophs–the eukaryote *Arabidopsis thaliana* [42], cyanobacteria *Synechocystis* sp. PCC 6803 [15,43], and the microalgae *Tisochrysis lutea* [44] and *Chlamydomonas reinhardtii* [45].

The biomass equation in metabolic modeling serves to drain metabolites (such as nucleotide triphosphates, amino acids, and carbohydrates) in their physiological ratios. The stoichiometric coefficients of biomass constituents are scaled such that flux through this reaction equals the exponential growth rate of the organism. Consequently, a biological fidelity test for any metabolic network model is to ensure that it is able to synthesize all biomass precursors. This constituted the next step of network curation implemented in this work. As the biochemical composition of *C. tobin* has not been measured experimentally, we assumed that it has the same biomass composition as that of *C. reinhardtii* (a well-studied microalga [45]). As UCYN-A remains uncultured under laboratory conditions, its exact biochemical composition is also unknown. Thus, to ensure that the constructed metabolic network replicates UCYN-A's metabolism accurately by synthesizing all necessary biomass precursors, we assembled a list of putative biomass constituents from existing genome-scale reconstructions of *Cyanothece* sp. ATCC 51142 [15] and *P. marinus* [16] (S2 Table). To achieve a comprehensive description of UCYN-A metabolism, the union set of precursors from both organisms was taken, barring phycocyanobilins and TCA cycle intermediates as the UCYN-A genome lacks genes encoding their synthesis. A list of essential genes and the corresponding blocked biomass precursors can be found in Table 3 in S1 Text.

## Modeling symbiosis between UCYN-A and its host

A gapfill procedure [46] was employed to concurrently restore biomass productivity in both UCYN-A and *C. tobin* using the two constructed GSMs. This yielded the minimal set of reactions which need to be added to both the metabolic networks to enable biomass production. Carbon was supplied as $CO_2$ to *C. tobin*, molecular nitrogen to UCYN-A, and a minimal medium used for all simulations. All reactions thus found were added to the respective metabolic networks after determining that the corresponding genes are present in that organism's genome (using a protein BLAST) (see S3 Table and S4 Table).

Next, we determined the minimal set of metabolite exchanges occurring between UCYN-A and its host. Let *I* be the set of all metabolites and *J* the set of all reactions present in the

symbiotic model. Matrix with elements $S_{ij}$ denotes the stoichiometry of metabolite $i$ in reaction $j$. The flux $v_j$ through every reaction $j$ was constrained to lie between an upper ($UB_j$) and lower ($LB_j$) bound. Feasible reaction directions were determined using the standard Gibbs free energy of change [47]. Metabolic networks of the two organisms were linked using transport reactions. Subsets $J_{transfer,UCYN}$ contains transfers from UCYN-A to the host and subset $J_{transfer,Host}$ denotes transfers from the host to UCYN-A. Binary variables $y_j$ were associated with subsets $J_{transfer,UCYN}$ and $J_{transfer,Host}$.

$$y_j = \begin{cases} 1, & \text{if transfer reaction } j \text{ carries flux} \\ 0, & \text{otherwise} \end{cases}$$

Experimentally-measured rates of carbon and nitrogen exchange were implemented as model constraints, wherein at most 17% of the fixed carbon (from the host) and at least 95% of the UCYN-A fixed amount of nitrogen was allowed to be exchanged between the organisms [8]. To this end, we defined parameters $N_{c,j}$ and $N_{N,j}$ to record the number of carbon and nitrogen molecules, respectively, that are present in a metabolite associated with the transfer reaction $j$. The total amount of nitrogen fixed was set by the flux through the $N_2$ fixation reaction (i.e., $v_{N_2fixation}$) and the amount of carbon fixed by the host by the flux through the carbon dioxide uptake reaction (i.e., $v_{CO_2\ exchange}$). Similar to a previous FBA study of diazotrophy for the marine cyanobacterium *Trichodesmium erythraeum* [48], carbon dioxide was the sole carbon source with a basis of 100 mmol gDW$^{-1}$ hr$^{-1}$ provided to the system.

As the exact ratio of biomass constituents of UCYN-A remain unknown, a sink reaction was defined for every biomass precursor (denoted by the set $J_{biomass,UCYN}$) and its lower bound set to $\varepsilon = 0.01$ mmol gDW$^{-1}$hr$^{-1}$ to ensure its production. The following mixed-integer linear program (MILP) minTransfers was solved to determine the minimal set of metabolites exchanged in the UCYN-A and host symbiosis to facilitate UCYN-A biomass production.

$$\text{minimize} \sum_{j \in (J_{transfer,UCYN} \cup J_{transfer,Host})} y_j \quad (\text{minTransfers})$$

s.t.

$$\sum_{j \in J} S_{ij}v_j = 0, \qquad \forall i \in I$$

$$\sum_{j \in J_{trans,Host}} N_{c,j}v_j \leq (0.17)(v_{CO_2\ exchange})$$

$$\sum_{j \in J_{trans,UCYN}} N_{N,j}v_j \geq 2(0.95)(v_{N_2fixation})$$

$$0 \leq v_j \leq y_j M, \qquad \forall j \in J_{trans,UCYN} \cup J_{trans,Host}$$

$$v_j \geq \varepsilon, \qquad \forall j \in J_{biomass,UCYN}$$

$$v_{CO_2\ Exchange} \leq 100,$$

$$LB_j \leq v_j \leq UB_j, \qquad \forall j \in J \setminus (J_{trans,UCYN} \cup J_{trans,Host})$$

$$v_j \in \mathbb{R}; \ y_j \in \{0,1\}, \quad \forall j \in \boldsymbol{J}$$

The first constraint imposes the pseudo steady-state mass balance constraint on every metabolite $i$. The next two constraints impose experimentally-measured rates of carbon and nitrogen exchange [8]. The number of active metabolite transfer reactions is recorded by forcing the associated binary variable $y_j$ to assume a value of one when the reaction carries flux. Every transfer reaction was constrained to be in the forward direction. Constant $M$ is large enough so as to ensure unconstrained flux through the transfer reaction (taken to be 1,000 in the current simulations). We also ensure that every UCYN-A biomass precursor can be synthesized by the combined host and UCYN-A metabolic networks at an $\varepsilon$ value. Carbon was supplied as $CO_2$ to the system and its net uptake constrained to be 100 mmol gDW$^{-1}$ hr$^{-1}$. Finally, every metabolic reaction was constrained to lie between a lower $LB_j$ and upper $UB_j$ bound.

Alternate trophic scenarios (S1 Table) were generated using integer cuts which (i) disallow previously identified solutions, and (ii) search for alternate optimal and sub-optimal solutions to explore all possible metabolite exchange scenarios. They are implemented by appending the following constraint to each subsequent MILP:

$$\sum_{j|y_j^k=1} y_j \leq \sum_j y_j - 1, \quad \forall k = 1, \ldots, K$$

where $k = 1,\ldots,K$ is the set of previously identified solutions. The value of $K$ was taken to be 100 in the current simulations.

The resultant flux distribution was computed using parsimonious flux balance analysis [49] following each trophic scenario. For generating the results shown in Figs 3 and 4, a variation of the above problem was solved which maximized the total UCYN-A nitrogen fixation flux (maxFixation):

$$\text{maximize } v_{N_2fixation} \ (\text{maxFixation})$$

s.t.

$$\sum_{j \in \boldsymbol{J}} S_{ij} v_j = 0, \quad \forall i \in \boldsymbol{I}$$

$$\sum_{j \in \boldsymbol{J}_{trans,Host}} N_{c,j} v_j \leq (0.17)(v_{CO_2 \ exchange})$$

$$\sum_{j \in \boldsymbol{J}_{trans,UCYN}} N_{N,j} v_j \geq 2(0.95)(v_{N_2fixation})$$

$$0 \leq v_j \leq y_j M, \quad \forall j \in \boldsymbol{J}_{trans,UCYN} \cup \boldsymbol{J}_{trans,Host}$$

$$v_{CO_2 \ Exchange} \leq 10,$$

$$LB_j \leq v_j \leq UB_j, \quad \forall j \in \boldsymbol{J} \setminus (\boldsymbol{J}_{trans,UCYN} \cup \boldsymbol{J}_{trans,Host})$$

$$v_j \in \mathbb{R}; \ y_j \in \{0,1\}, \quad \forall j \in \boldsymbol{J}$$

This was followed by determining the minimal number of metabolites exchanged (using minTransfers) while constraining the nitrogen fixation flux to be its maximum determined value using maxFixation. For comparing between oxygen scavenging mechanisms, an additional constraint was added to maxFixation wherein the UCYN-A oxygen uptake rate was set to be 0.1 mmol gDW$^{-1}$ hr$^{-1}$.

The General Algebraic Modeling System (GAMS) (using the Cplex solver) was used to conduct constraints-based analysis and Python 2.7 used to generate all input files and analyze results. All computations were carried out on dual 10-core and 12-core Intel Xeon E5-2680 and Intel Xeon E7-4830 quad 10-core processors that are part of the Institute for Computational and Data Sciences Advanced Cyber Infrastructure (ICDS-ACI) cluster of High-Performance Computing Group of Pennsylvania State University.

## Supporting information

**S1 Table. List of metabolites transferred from the algal host to UCYN-A to facilitate biomass synthesis and nitrogen fixation (summarized from 100 alternate scenarios).**
(XLSX)

**S2 Table. List of UCYN-A biomass precursors derived from metabolic reconstructions of** *Cyanothece* **sp. ATCC 51142 and** *P. marinus.*
(XLSX)

**S3 Table. Genome-scale metabolic reconstruction of UCYN-A.**
(XLSX)

**S4 Table. Genome-scale metabolic reconstruction of** *C. tobin.*
(XLSX)

**S1 Text. Summarizes GSM statistics, and additional calculations such as impact of fixed nitrogen availability on the symbiosis, and essentiality and metabolic control analysis in UCYN-A.**
(DOCX)

## Author Contributions

**Conceptualization:** Himadri B. Pakrasi, Jonathan P. Zehr, Costas D. Maranas.

**Data curation:** Debolina Sarkar, Marine Landa.

**Formal analysis:** Debolina Sarkar, Marine Landa, Anindita Bandyopadhyay, Costas D. Maranas.

**Funding acquisition:** Himadri B. Pakrasi, Jonathan P. Zehr, Costas D. Maranas.

**Investigation:** Debolina Sarkar, Marine Landa.

**Methodology:** Debolina Sarkar, Costas D. Maranas.

**Project administration:** Himadri B. Pakrasi, Jonathan P. Zehr, Costas D. Maranas.

**Resources:** Costas D. Maranas.

**Software:** Debolina Sarkar, Costas D. Maranas.

**Supervision:** Himadri B. Pakrasi, Jonathan P. Zehr, Costas D. Maranas.

**Validation:** Debolina Sarkar, Marine Landa, Anindita Bandyopadhyay.

**Visualization:** Debolina Sarkar.

**Writing – original draft:** Debolina Sarkar, Costas D. Maranas.

**Writing – review & editing:** Debolina Sarkar, Marine Landa, Anindita Bandyopadhyay, Himadri B. Pakrasi, Jonathan P. Zehr, Costas D. Maranas.

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
