## [Decision Letter · Decision Letter 0]

14 Dec 2020

Dear Dr. Maranas,

Thank you very much for submitting your manuscript "Elucidation of trophic interactions in an unusual single-cell nitrogen-fixing symbiosis using metabolic modeling" for consideration at PLOS Computational Biology.

As with all papers reviewed by the journal, your manuscript was reviewed by members of the editorial board and by several independent reviewers. In light of the reviews (below this email), we would like to invite the resubmission of a significantly-revised version that takes into account the reviewers' comments.

We cannot make any decision about publication until we have seen the revised manuscript and your response to the reviewers' comments. Your revised manuscript is also likely to be sent to reviewers for further evaluation.

Sincerely,

Vassily Hatzimanikatis

Associate Editor

PLOS Computational Biology

Douglas Lauffenburger

Deputy Editor

PLOS Computational Biology

Reviewer's Responses to Questions

**Comments to the Authors:**

Reviewer #1: I have read the manuscript entitled "Elucidation of trophic interactions in an unusual single-cell nitrogen-fixing symbiosis using metabolic modeling" by Sarkar et al. submitted for publication to Plos Computational Biology. The manuscript deals with the metabolic reconstruction and modeling of the trophic interactions in a single-cell nitrogen fixing symbiosis, the one established by UCYN-A and a halophyte algal host.

In general, it feels like the reconstruction was properly performed (despite a lack of details on the actual implementation of the model, see below) but, on the other end, other additional aspects could have been investigated about this interesting biological system using the integrated model.

This system is characterized by a tremendous lack of information, mainly due to the fact that by both UCYN-A and its host are uncultured. The work done by the authors is profoundly influenced by this as both the reconstructions were built using other (more or less) related organisms as template.

In this context, it seems that the authors made all the efforts to include the available annotations (C. Parka and B. Bigelowii are said to be "partially sequenced". Btw, what does this mean? Does it mean that only a small fraction of the genome is actually sequenced or that the whole sequence is available as draft?) of the host and the symbiont to reconstruct a model that is as close as possible to the actual metabolic network of the two. It would be interesting to know more details about the two reconstructions and a possible guess on how complete they actually are. For example, how many genes of the two organisms were actually included in the corresponding reconstructions? How many transport reactions were included in the reconstructions? A few more tables including this information would facilitate the reading.

I am deeply concerned by the fact that the authors did not provide (as additional material or as link to a public repository) the models and the codes they used to run the simulations. They only provided the models in xls format which is not the standard in genome scale metabolic modeling. I was expecting sbml files or something comparable. The description of the implementation of the model is restricted to lines 459-464 which do not provide enough details/material to reproduce the results or apply the proposed approach to other study-cases.

The authors have used constrained-based metabolic modeling to infer the interconnection between these two organisms and the minimal set of nutrients that are probably exchanged by the two organisms to reciprocally sustain life. In this "do ut des" system (and in many other symbioses), trade-offs play an interesting role. In this case, the host needs to invest resources to have usable nitrogen back from the symbiont and viceversa. But the question is, which is the limit of the system? Is there a point behind which the system collapses because too many (or too few) resources have been invested by the host (or by the symbiont)? This aspect has been overlooked by the authors but something in the range of a Pareto frontier analysis could highlight interesting features of the system.

Additionally, the effect of a number of external perturbations on the system could have been explored. I am listing here a few of them but the list can surely be expanded by the authors.

1. What would happen if the host could retrieve fixed nitrogen from the external environment? Would this be possible in a real case scenario? How would this impact the metabolic cross-talk of the two organisms?

2. There is no investigation on the role of specific genes in the establishment and maintenance of the symbiosis. A gene-essentially analysis (for example in different environmental conditions, as suggested above) might have revealed interesting insights.

3. A similar analysis could be performed at the reaction-level, for example constraining the flux in all the reactions of the model (one at the time, obviously) to higher or lower values in respect to the reference FBA flux distribution.

4. A deeper analysis of the combined effects of CO2 and light on the N2-fixing symbiont should be performed. Is nitrogen fixation influenced by light intensity? Constraining light adsorption at different values and checking what happens I the symbiont model could reveal interesting insights.

5. To put the work in perspective the results obtained with this system could be discussed in relation to other similar (but unrelated) symbioses (e.g. the one between plant and bacteria, Medicago and Sinorhizobium).

Reviewer #2: Overall, this is an interesting paper that explores potential trophic interactions between two uncultured species, with a symbiotic nitrogen fixation and carbon fixation relationship. Although there really aren’t many new computational methods in this paper, it does a nice job of showing how community models can be used to explore these trophic interactions, and the carbon transfer/nitrogen transfer constraints add some novelty, as does the exploration of limits for different O2 scavenging processes.

I have only a few minor comments and suggestions for this paper:

Comments:

1.) The description of the biomass in the results seems to imply all of the coefficients are equal and minimal in the UCYN-A model:

“FBA was carried out by requiring that each UCYN-A biomass precursor was produced at a minimal level (as the exact biomass composition is unknown)”

But the methods say the biomass composition from C. reinhardtii was used. Perhaps the sentence in the results could be clarified?

2.) When enforcing boolean matching of the GPR during reconstruction of UCYN-A, how many partial matches were rejected? Did gap filling add reactions associated with partial matches back into the model? Could it be possible UCYN-A has some alternative protein complex stoichiometry?

3.) It seems like some model stats in the results would be helpful. The number of reactions. The number of genes. The amount of gap filling required.

4.) The authors make a significant point that UCYN-A is not cultured, yet they do have a genome for annotation and reconstruction. It seems like they should describe the provenance of this genome. Citations are made, but some minimal information on this in the paper itself (a few sentences) would be helpful. Is the genome closed? Given it’s minimized nature, errors in assembly or binning could be significant.

5.) It’s not totally clear from the results. Does UCYN-A have all the enzyme components for both oxygen scavenging system explored?

6.) It seems like the oxygen scavenging comparison should include the costs of dealing with the H2O2 byproduct of the Mehler reaction. It’s mentioned in the text that systems exist to deal with this byproduct, but what are the metabolic costs?

7.) Perhaps in figure 4 the distinctive reactions in each mechanism could be highlighted with a different color to help clarify this to the reader?

**Have all data underlying the figures and results presented in the manuscript been provided?**

Reviewer #1: **No: **Codes and metabolic reconstructions in a computable form are missing

Reviewer #2: Yes

PLOS authors have the option to publish the peer review history of their article (what does this mean?). If published, this will include your full peer review and any attached files.

Reviewer #1: **Yes: **Marco Fondi

Reviewer #2: No
---

## [Decision Letter · Decision Letter 1]

20 Apr 2021

Dear Dr. Maranas,

We are pleased to inform you that your manuscript 'Elucidation of trophic interactions in an unusual single-cell nitrogen-fixing symbiosis using metabolic modeling' has been provisionally accepted for publication in PLOS Computational Biology.

Best regards,

Vassily Hatzimanikatis

Associate Editor

PLOS Computational Biology

Douglas Lauffenburger

Deputy Editor

PLOS Computational Biology

Reviewer's Responses to Questions

**Comments to the Authors:**

Reviewer #1: All the points I have raised in my previous review have been addressed

Reviewer #2: The authors have responded well to all of my comments. The paper appears to be ready for publication in my view.

**Have all data underlying the figures and results presented in the manuscript been provided?**

Reviewer #1: Yes

Reviewer #2: Yes

PLOS authors have the option to publish the peer review history of their article (what does this mean?). If published, this will include your full peer review and any attached files.

Reviewer #1: **Yes: **Marco Fondi

Reviewer #2: No

---

## [Editor Report · Acceptance letter]

29 Apr 2021

PCOMPBIOL-D-20-01569R1 

Elucidation of trophic interactions in an unusual single-cell nitrogen-fixing symbiosis using metabolic modeling

Dear Dr Maranas,

I am pleased to inform you that your manuscript has been formally accepted for publication in PLOS Computational Biology. Your manuscript is now with our production department and you will be notified of the publication date in due course.

With kind regards,

Katalin Szabo
